# Regional differences and dynamic evolution of agricultural water resources utilization efficiency in China

Qian Zeng, Shuya Cao[ID]*, Jiayi H. E.

School of Economics and Finance, Xi'an International Studies University, Xi'an, Shaanxi, China

* caoshuya2022@163.com

**Data Availability Statement:** All data are present within the manuscript and its Supporting information files.

**Funding:** This research was funded by General Project of Shaanxi Provincial Department of

## Abstract

Improving water resources utilization efficiency is conducive to achieving the sustainable development of water resources. It is essential to explore the regional differences and dynamic evolution of agricultural water resources utilization efficiency in China to promote high-quality development of agriculture. In this study, based on the unexpected output, we build a super slack-based measure model to measure agricultural water resources utilization efficiency in China's provinces from 2007 to 2018. In addition, we use the Dagum Gini coefficient to analyze the source of regional differences. Finally, we construct the distributed dynamics model to explore the distribution of the dynamic evolution trend of China's agricultural water resources utilization efficiency. The results reveal that regional difference is the main source of the overall difference in agricultural water resources utilization efficiency in China. Spatial imbalance exists in the development of agricultural water resources utilization efficiency in China. The agricultural water resources utilization efficiency in various provinces and regions of China is relatively stable, having the characteristics of club convergence. The probability of maintaining the initial state is high, and the internal mobility is low. However, with time, the degree of club convergence decreases.

## Introduction

Resources are scarce. Water is not only an important resource for human survival and development, but also an important component of the ecological environment. The sustainable development of water resources is related to the global sustainable development. At the global level, population growth, destruction of the ecological environment, water pollution and improper management of water resources have exacerbated the shortage of water resources. At the same time, for fast-growing economies, the increase in demand for water resources caused by the growth of industrialization will threaten the supply of water resources, such as China.

At present, China is in the stage of rapid development, and constantly promote economic and social adjustment. The accelerated process of industrialization and urbanization has produced great pressure on limited water resources. At the same time, in order to ensure agricultural production and meet the food demand of 1.4 billion people in China, it is necessary to

Science and Technology Soft Science Research Program ' Study on the Impact of Environmental Regulation on Green Technology Innovation in Shaanxi Province ' ( 2022KRM116 ), Shaanxi Social Science Fund Project ' Research on the Governance Path of Rural Water Pollution in Shaanxi Province under the Rural Revitalization Strategy ' ( 2020D025 ), Research on the driving policy of green technology innovation of enterprises in Xi ' an under the key projects ' and ' double carbon ' of Xi'an Social Science Planning Fund ' ( 22JX137 ) and Key Research Project of Xi'an International Studies University ' Governance Path and Emission Reduction Evaluation of Rural Water Pollution under Rural Revitalization Strategy ' ( 19XWA03 ).The funders had no role in study design, data collection and analysis, decision to publish, or preparation of the manuscript.

**Competing interests:** The authors declare no conflict of interest.

improve water resources utilization efficiency, reduce agricultural water consumption, and obtain more output with less input.

The spatial and temporal distribution of water resources in China is uneven, the demand for agricultural water resources is huge, the pollution problem is serious, and the regional differences (eastern, central and western) are very obvious. When studying the utilization efficiency of agricultural water resources, many studies focus on the irrigation water use efficiency and regional distribution of the planting industry [1–5], and put forward reasonable suggestions, such as optimizing irrigation technology and developing water-saving agriculture [5, 6]. There are fuzzy comprehensive evaluation method [7, 8], analytic hierarchy process [8, 9], stochastic frontier method and data envelopment analysis (DEA) [6, 10–14] in the evaluation method of agricultural water resource utilization efficiency. With the deepening of DEA research, the use of DEA method to measure the utilization efficiency of agricultural water resources has gradually changed from the original standard efficiency DEA model to the super efficiency DEA model [15–17], from only expected output model to include undesirable output [18, 19], and SBM model can consider the relaxation variable problem [20, 21], which has more advantages in efficiency measurement, so it has been more widely used [11, 14–17, 20, 22, 23]. Most of the economic studies on agricultural water resource utilization efficiency are analyzed from the influencing factors of efficiency changes [14–16, 24–26]. Some studies also include pollution emissions in the calculation of agricultural water resource utilization efficiency, that is, under the constraint of pollu-tion emissions, to measure agricultural water resource utilization efficiency [14, 27]

From the perspective of the research scope of agricultural water resources utilization efficiency, most scholars study the changes of agricultural water resources utilization efficiency in various provinces and cities nationwide, and some study the agricultural water resources utilization efficiency in a certain region, such as Heilongjiang Province, Shandong Province, Northwest inland river basin, Yellow River basin, provinces and cities along the Yangtze River basin and Beijing-Tianjin-Hebei region and so on. At the same time, most studies focus on the regional differences [17, 28], spatial heterogeneity and spatio-temporal convergence and divergence [28–31] in the utilization efficiency of agricultural water resources, but there are relatively few studies on the sources and dynamic evolution of regional differences in the utilization efficiency of agricultural water resources.

Existing research has laid a good foundation for this article, providing very valuable theoretical and methodological support, but to be further improved. In this study, we used agricultural, forestry, animal husbandry and fishery data from 30 provinces (excluding Tibet, Hong Kong, Macao and Taiwan) in China (hereinafter referred to as agriculture). Since the traditional DEA model cannot distinguish multiple decision-making units, we chose an super-efficient SBM (Slacks-Based Measure) model. The differences of agricultural water use status at provincial level in China from 2007 to 2018 were quantitatively analyzed. Comparing to previous reseaches on Agricultural Water Resources Utilization Efficiency [12–14, 20], the innovation of this paper is to use a variety of pollutants as undesirable outputs and use the Dagum Gini coefficient to analyze Agricultural Water Resources Utilization Efficiency on a measured basis. This method can describes the size and source of regional differences in the eastern, central and western regions, and ef-fectively solve the problem of cross-overlap between samples as well. Moreover, Kernel density estimation in distribution dynamics can be used to analyze distribution location, shape and extensibility [32], contributing to the study of the overall morphology and changes of Agricultural Water Resources Utilization Efficiency. However, this method is difficult to reflect the relative position changes and their likelihood that are distributed across regions in the course of changes in Agricultural Water Resources Utilization Efficiency, while the Markov chain method can focus on solving the possibility of state transition

in each region during the change of Agricultural Water Resources Utilization Efficiency, and can reflect the internal dynamics of the distribution of Agricultural Water Resources Utilization Efficiency greatly. The kernel density estimation and Markov chain method can complement each other in the analysis of the dynamic evolution of the distribution of Agricultural Water Resources Utilization Efficiency, and reflect the dynamic evolution of the distribution in the process of the change of Agricultural Water Resources Utilization Efficiency in China more deeply.

## Materials and research methods

### Study area

China's total water resources are abundant. In 2018, China's total water resources are 274.625 billion cubic meters, and the per capita water resources are only 1957.7 cubic meters, ranking 109th in the world. The per capita water resources are scarce. In 2018, China's total water consumption was 601.55 billion cubic meters, of which agricultural water consumption was 369.31 billion cubic meters, accounting for 61.4 percent of the total water consumption in the country. Agricultural water consumption accounts for a large proportion. However, due to the influence of geographical location and climatic conditions, the distribution of agricultural water resources is uneven. Agricultural development in water-deficient areas is facing the dilemma of water shortage. At the same time, water-rich areas are facing serious waste of water resources and low water resources management.

S1 Fig, Fig 1 shows the distribution map of agricultural water consumption by province in China in 2018, which is divided by natural breakpoint method. Agricultural water includes farmland irrigation water (divided into paddy field, irrigation land, vegetable field, vegetable field, groundwater development), forest, animal husbandry, forage, livestock (divided into fruit, grassland, pond, animal husbandry, groundwater development), domestic water for rural residents and water for rural ecological environment. Previous studies mainly focused on the irrigation water consumption of planting industry at provincial or regional levels (such as the eastern, central and western regions). This paper considers water resources utilization efficiency in 30 provinces of China (some indicators cannot be obtained from Tibet data), and divides them into three regions : the east, the middle and the west according to geographical location, economic development and national development planning. The root causes of the differences in water resources utilization efficiency are explored, and the dynamic evolution trend of China's agricultural water resources utilization efficiency from 2007 to 2018 is analyzed by nuclear density estimation. At the same time, the Markov chain is used to analyze the evolution trend of different levels of China's agricultural water resources utilization efficiency in the next 1,2,3 years.

### Research methods

**Super-efficiency SBM model based on unexpected output.**    In this study, the agricultural water resources utilization efficiency (under the convex frontier) is calculated by using the super-effciency slacks-based measure (Super-SBM) model.

Charnes was the first to propose a new method for measuring efficiency data envelopment analysis(DEA) [33]. The traditional DEA method does not need to set the specific form of the function, so it is favored by many scholars, but it ignores the relaxation of variables. When the efficiency values of decision-making units are all 1, we cannot compare the magnitude of their efficiency. The efficiency of the super-efficiency DEA model can be greater than 1, that is, the efficiency levels of all decision-making units can be compared.

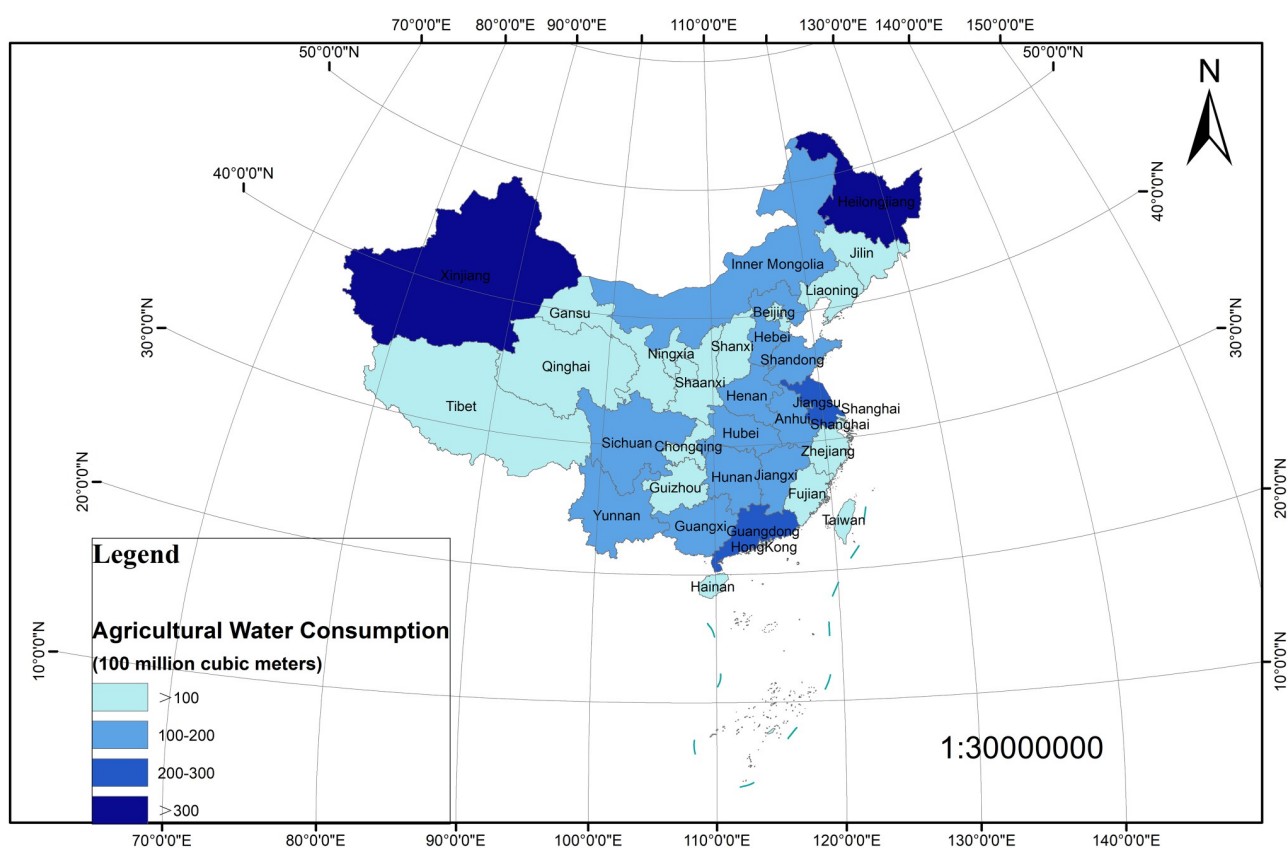

**Fig 1. Agricultural water consumption in China by province for 2018.** Note:Map created using ArcGIS[10.7]. SHP data downloaded from standard map service system(http://www.resdc.cn/).

Tone proposed the SBM model in 2001 to solve the problem of slack variables [21, 34]. SBM model is a non-radial and non-angular DEA analysis method based on slack variable measures. Undesirable output-based SBM models address slack in input and output variables, thereby minimizing bias in initial measurements. The super-efficiency SBM model combines the super-efficiency DEA model and the SBM model. It is also a method based on data envelopment analysis, which measures the effectiveness of all decision units and the slackness of input and output variables.

Suppose there are n decision-making units, each with m inputs, expected output $r_1$, and unexpected output $r_2$. Let $X(X \subset R^m)$, $Y^e(Y^e \subset R^{r_1})$ and $(Y^u \subset R^{r_2})$ be matrices, such that $X = [x_1, \ldots, x_n] \in R^{m*n}$ and $Y = [y_1^e, \ldots, y_n^e] \in R^{r_1*n}$. Te form of the super-efficiency SBM model is as follows:

$$min = \frac{\frac{1}{m}\sum_{i=1}^{m}\frac{\overline{x}}{x_{ik}}}{1/(r_1 + r_2)*(\sum_{r=1}^{r_1}\frac{\overline{y^e}}{y_{rk}^e} + \sum_{q=1}^{r_2}\frac{\overline{y^u}}{y_{uk}^u})} \quad (1)$$

Among them,

$$
\begin{cases}
\overline{x} \geq \sum_{j=1 \neq k}^{n} x_{ij}\lambda_{j}, i = 1, \ldots, m \\
\overline{y^{e}} \leq \sum_{j=1 \neq k}^{n} y_{rj}^{e}\lambda_{j}, i = 1, \ldots, r_{1} \\
\overline{y^{e}} \leq \sum_{j=1 \neq k}^{n} y_{dj}^{u}\lambda_{j}, q = 1, \ldots, r_{2} \\
\lambda_{y} \geq 0, j = 1, \ldots, n; j \neq 0 \\
\overline{x} \geq x_{k}, k = 1, \ldots, m \\
\overline{y^{e}} \leq y_{k}^{e}, e = 1, \ldots, r_{1} \\
y^{u} \geq y_{k}^{u}, u = 1, \ldots, r_{2}
\end{cases}
\tag{2}
$$

Based on the Super-SBM model (Eq 1) and its constraint formula, the agricultural water resources utilization efficiency for the different provinces was calculated for the period 2007–2018 using Maxdea 8 ultra software.

**Dagum's Gini coefficient.** Dagum's Gini coefficient and its decomposition are used to measure regional differences. The Gini coefficient is decomposed into the following three parts according to the subgroup decomposition method: intra-regional gap, inter-regional gap, and hypervariable density [35–38]. This method is used to divide 30 provinces into eastern, central, and western regions. Eastern Region: Beijing; Tianjin; Hebei; Liaoning; Shanghai; Jiangsu; Zhejiang; Fujian; Shandong; Guangdong; Hainan Central Region: Shanxi; Jilin; Heilongjiang; Anhui; Jiangxi; Henan; Hubei; Hunan Western Region: Inner Mongolia; Guangxi; Sichuan; Guizhou; Yunnan; Xizang; Shaanxi; Gansu; Qinghai; Ningxia; Xinjiang Tibet data not available. The definition of the overall Gini coefficient is presented in Eq (3), where k denotes the number of regions divided; n denotes the number of provinces; $Y_{ji}(Y_{hr})$ denotes the agricultural water resources utilization efficiency of province i(r) in region j(h); nj(nk) denotes the number of provinces in region j(h), and Y denotes the mean value of water resources utilization efficiency of all provinces.

$$
G = \frac{\sum_{j=1}^{k} \sum_{h=1}^{k} \sum_{i=1}^{n_{j}} \sum_{r=1}^{n_{h}} |Y_{ji} - Y_{hr}|}{2n^{2}\overline{Y}}
\tag{3}
$$

$$
G_{jj} = \frac{\frac{1}{2\overline{Y}_{j}} \sum_{i=1}^{n_{j}} \sum_{r=1}^{n_{j}} |Y_{ji} - Y_{jr}|}{n_{j}^{2}}
\tag{4}
$$

$$
G_{jh} = \frac{\sum_{i=1}^{n_{j}} \sum_{r=1}^{n_{h}} |Y_{ji} - Y_{hr}|}{n_{j}n_{h}(\overline{Y}_{j} + \overline{Y}_{h})}
\tag{5}
$$

Formulas (4) and (5) denote the Gini coefficient $G_{jj}$ in region j and the Gini coefficient $G_{jh}$ between regions j and h, respectively. $(Y_{j})(Y_{h})$ represents the average water resources utilization efficiency in region j(h).

The following variables are defined as follows:

$$P_j = n_j/n \tag{6}$$

$$S_j = n_j\overline{Y}_j/n\overline{Y} \tag{7}$$

$$M_{jh} = \int_0^\infty dF_j(Y) \int_0^Y (Y-x)dF_h(x) \tag{8}$$

$$N_{jh} = \int_0^\infty dF_h(Y) \int_0^Y (Y-x)dF_j(x) \tag{9}$$

$$D_{jh} = \frac{M_{jh} - N_{jh}}{M_{jh} + N_{jh}} \tag{10}$$

$D_{jh}$ represents the relative impact of agricultural water resources utilization efficiency between regions j and h; $M_{jh}$ represents the difference in agricultural water resources utilization efficiency between regions j and h; all $Y_{ji} - Y_{hr} > 0$ sample values are equal to the total mathematical expectation; $N_{jh}$ represents the first-order moment of super variation, which is the mathematical expectation of the sum of all the sample values of $Y_{hr} - Y_{ji} > 0$; function $F_j$ ($F_h$) represents the cumulative density distribution function in region $j(h)$.

The intra-regional gap $G_w$, the inter-regional gap $G_{nb}$, and the hypervariable density $G_t$ are expressed as

$$G_w = \sum_{j=1}^k G_{jj}P_jS_j \quad G_{nb} = \sum_{j=2}^k \sum_{h=1}^{j-1} G_{jh}D_{jh}(P_jS_h + P_hS_j) \tag{11}$$

$$G_{nb} = \sum_{j=2}^k \sum_{h=1}^{j-1} G_{jh}D_{jh}(P_jS_h + P_hS_j) \tag{12}$$

$$G_t = \sum_{j=2}^k \sum_{h=1}^{j-1} G_{jh}(P_jS_h + P_hS_j)(1 - D_{jh}) \tag{13}$$

**Kernel density estimation.** Kernel density estimation is a type of nonparametric estimation that is often used in the study of unbalanced distribution [32, 39]. Assuming $f(x)$ is a density function of random variable $X$, the probability density at point x is estimated using Formula (14), where $K()$ is the kernel function; $N$ is the number of observations; $X_i$ is the independent and identically distributed observation; x is the mean value, and h is the bandwidth. The larger the bandwidth, the smoother the estimated density function curve and the lower the estimation accuracy. Conversely, the smaller the bandwidth, the smoother the estimated density function curve and the higher the estimation accuracy. In this study, Gaussian kernel estimation is used to estimate the dynamic evolution of the distribution of agricultural water

resources utilization efficiency in China, as is shown in Formula (15).

$$f(x) = \frac{1}{Nh}\sum_{i=1}^{N}K\left(\frac{X_i - x}{h}\right) \tag{14}$$

$$K(x) = \frac{1}{\sqrt{2\pi}}\exp\left(-\frac{x^2}{2}\right) \tag{15}$$

**Analysis of Markov Chain.** We construct a Markov transition probability matrix to study the dynamic transfer trend of water resources utilization efficiency in various regions over time [2, 23, 24]. In this study, the transition probability matrix with a year period is constructed using MATLAB. Formula (16) is used to calculate the transition probability matrix, where $P_{ij}^{t,t+d}$ is the transfer probability of the agricultural water resources utilization efficiency of a certain region from type i in year t to type j in year t+d; $n_{ij}^{t,t+d}$ is the number of regions that belong to type i in year t after year d in the research period; $n_i^t$ is the number of regions that belong to type i in year t. If China's agricultural water resources utilization efficiency is divided into λ types, λ × λ transfer probability matrix can be constructed, and the dynamic evolution process of water resources utilization efficiency distribution in China's provinces can be explored from the perspective of transfer probability.

$$P_{ij}^{t,t+d} = P\{X_{t+d} = j \mid X_t = i\} = \frac{\sum_{t=2004+d}^{2018} n_{ij}^{t,t+d}}{\sum_{t=2004}^{2018-d} n_i^t} \tag{16}$$

## Variable selection and data source

This study takes 30 provinces in China as decision-making units (DMUs) and analyzes their panel data from 2007 to 2018. When using the DEA method to measure the index of agricultural water resources utilization efficiency, it is necessary to define input and output variables.

The selection of input–output factors to measure the utilization efciency of agricultural water resources follows the principles of availability and operability. The input variables included: (1) agricultural water consumption, (2) the number of employees in agriculture, forestry, animal husbandry, and fishery, (3) the total power of agricultural machinery, and (4) the expenditure of local fnance on agriculture, forestry, and water afairs. In terms of output, the added value in agriculture, forestry, animal husbandry, and fishery (based on 2007) was used as the expected output, while ammonia nitrogen emission, agricultural chemical oxygen demand emission, and agricultural carbon emission comprised the unexpected output [14, 20, 40].

The reason why we choose agricultural ammonia nitrogen emissions and agricultural chemical oxygen demand as undesirable outputs is that these agricultural pollutants lead to soil pollution and groundwater pollution [2, 20, 39, 40]. We choose agricultural carbon emissions as undesirable outputs because agricultural carbon emissions lead to rural environmental pollution.

## Regional differences in agricultural water resources utilization efficiency in China

### The calculation result of agricultural water resource utilization efficiency (based on super efficiency SBM-DEA model)

At the national level, the national average level of agricultural water resources utilization efficiency from 2007 to 2018 showed a continuous decline, followed by a slight upward trend. Compared with the early stage of the study, the national average level of water resources utilization efficiency in 2018 is still in a downward state. Among them, only water resources utilization efficiency in Beijing, Tianjin, Heilongjiang, Shanghai, Shandong, Guangdong, Chongqing, Sichuan, Guizhou and Ningxia increased compared with the initial period of the investigation, and water resources utilization efficiency in the provinces decreased compared with the initial period of the investigation, namely, 19 provinces, municipalities and autonomous regions in Hebei, Shanxi, Inner Mongolia, Liaoning, Jilin, Jiangsu, Anhui, Fujian, Jiangxi, Henan, Hubei, Hunan, Guangxi, Hainan, Yunnan, Shaanxi, Gansu, Qinghai and Xinjiang. Among them, water resources utilization efficiency in Beijing has changed greatly, with the efficiency difference of 2.11 in 11 years. The proportion of agricultural water consumption in Beijing's total water consumption is getting lower and lower, and the number of people engaged in agriculture, forestry, animal husbandry and fishery is decreasing year by year, which reflects that Beijing has given priority to the development of other industries. At the same time, in recent years, Beijing's ecological agriculture and leisure agriculture have developed rapidly, which is also conducive to the improvement of agricultural water resources utilization efficiency. Beijing exhibited a relatively large change in agricultural water resources utilization efficiency, having an efficiency difference of 2.11 in 11years. This is mainly due to Beijing's prioritization towards Secondary and tertiary industries, as highlighted by the annual decrease in the city's employment in agriculture, forestry, animal husbandry, and fishery. The annual agricultural water consumption in Beijing only accounts for less than 10 percent of the region's total water consumption. Shanghai maintained a slight upward trend during the study period, and the average utilization efficiency of agricultural water resources was above 1, which was similar to Beijing.

As can be seen from S2 Fig, Fig 2 Shandong, Hainan and other coastal areas of agricultural water advantage is superior to other places, highlighting the advantages of these areas in agriculture, technology and economy. In areas with less precipitation, relatively lagging agricultural water use technology and low management experience (such as Gansu, Shanxi, Jilin, Xinjiang, Anhui), the utilization level of water resources is low. In provinces with relatively large water resources such as Hubei, Anhui, Jiangxi and Hunan, agricultural irrigation technology and management system are undeveloped, resulting in the water resource efficiency level far below the national average. In Chongqing, Guangxi, Sichuan, Guizhou and Yunnan, rainfall is sufficient, but the utilization rate of water resources is low. Although their efficiency level is higher than other regions, but due to the overall level is low, the absolute difference is not so obvious.

### The regional difference of water resource utilization efficiency (based on Dagum's Gini coefficient))

First, the overall regional differences and evolution trend of water resources utilization efficiency in China. Fig 3a shows that the average Gini coefficient of water resources utilization efficiency in China is 0.228. From the perspective of evolution process, the overall regional gap of agricultural water resources utilization efficiency in China has shown a fluctuating upward

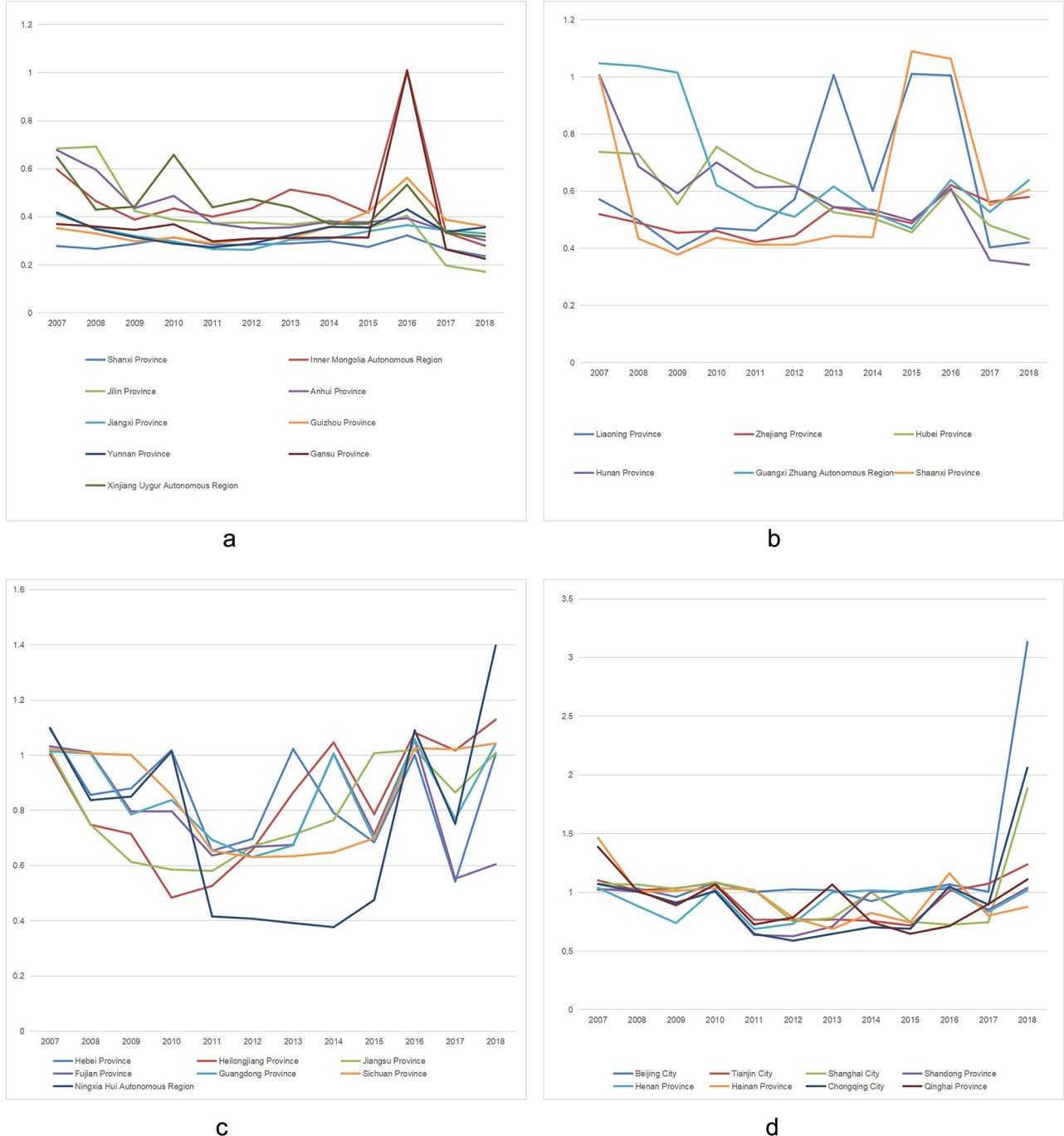

**Fig 2. Agricultural water resources utilization efficiency of 30 provinces, autonomous regions, and cities in China from 2007 to 2018.** The above four charts are listed seperately according to the average water use efficiency of (a) below 0.5, (b) 0.5–0.7, (c) 0.7–0.9, and (d) 0.9 and above.)

trend since 2007, reaching two small peaks in 2010 and 2014, reaching the minimum value of the overall regional gap in 2016. From 2016, the overall regional gap has increased significantly, reaching more than 0.3 in 2018. Overall, the overall regional gap of China's water resources utilization efficiency showed a trend of first decreasing, then increasing, then

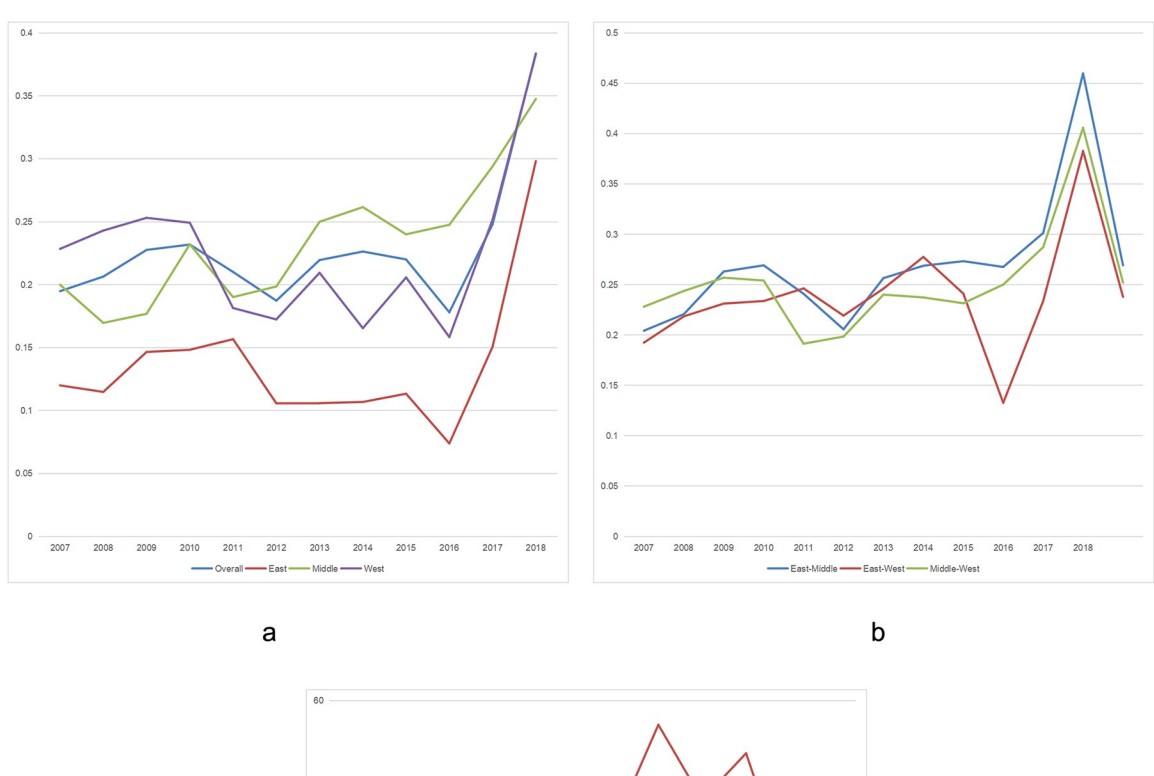

**Fig 3. Diagram of decomposition results of Dagum Gini coefficient.**

decreasing and finally increasing substantially, which increased by 8.8 percent compared with the initial period.

Second, the regional differences and evolution trend of agricultural water resources utilization efficiency in China. It can be seen from Fig 3a that the regional differences in water resources utilization efficiency in the three regions of eastern, central and western China are at different levels, but they show similar fluctuation trends. Throughout the investigation period, the average intra-regional disparity in the eastern region was significantly lower than that in

the central and western regions, which was 0.137; the mean value of the western region is significantly higher than that of the eastern region, which is 0.225; the central region has the largest average of 0.234. From the overall evolution trend, the overall evolution trend of the eastern, central and western regions is 'N' type, and there is a slight fluctuation in some years, reaching the bottom of the valley in 2016, reaching the lowest value of regional differences. After 2016, regional differences increased significantly, reaching the maximum in 2018. The intraregional differences in the western region were higher than those in the central region in 2007–2011 and higher in the central region than in the western region in 2011–2018. Compared with 2004, the intra-regional differences in the eastern, central and western regions increased by 13.5 percent, 6.7 percent and 6.2 percent respectively in 2018.

Third, the inter-regional differences in the agricultural water resources utilization efficiency in China and its evolution trend. Fig 3b shows that the gap between the central and eastern and western regions is large, and the difference between the eastern and western regions is relatively small. The average inter-regional disparities in the eastern, central and western regions were 0.269, 0.238 and 0.252, respectively. From the overall evolution trend, the change trend of regional differences in the eastern and central regions shows 'N' type. Compared with the early stage of the investigation, the regional differences in the eastern and central regions increased by 11.4 percent in 2018. In addition to a brief decline in individual years, the regional differences between the east and the west showed a fluctuating upward trend. Compared with the early stage of the investigation, the regional differences between the east and the west increased by about 9.0 percent in 2018. The difference between the central and western regions showed a slight decrease and then a slight increase, and then it increased significantly. Compared with the early stage of the investigation, the difference between the central and western regions increased by 7.1 percent.

Fourth, the contribution rate of regional differences in agricultural water resources utilization efficiency in China. It can be seen from Fig 3c that during the entire investigation period, the contribution rate of inter-regional disparity is the largest, and the contribution of intra-regional disparity and hypervariable density are almost the same. In terms of the overall evolution trend, the contribution rate of inter-regional disparity shows a fluctuating upward trend. Compared with 34.8 percent in the initial period of the investigation, the contribution rate of inter-regional disparity in 2018 is 45.3 percent, with an increase of 2.7 percent. The contribution rate of intra-regional differences shows a slight downward trend, and the overall fluctuation is not obvious, showing a 'V' type change trend; the excess density showed a fluctuating downward trend. Compared with the initial period, the contribution rate of excess density in 2018 decreased by 2.7 percent.

## The distribution dynamic evolution of agricultural water resources utilization efficiency in China

### Kernel density estimation of agricultural water resources Utilization efficiency))

(1) Kernel density estimation of agricultural water resources utilization efficiency in 30 provinces of China. In this paper, the kernel density of agricultural water resources utilization efficiency in 30 provinces in China is estimated by using the Gaussian kernel function with matlab software, and the dynamic evolution law of agricultural water resources utilization efficiency is described from the time dimension, as shown in Fig 4a. It can be found that : first, during the whole investigation period, the distribution of agricultural water resources utilization efficiency in 30 provinces in China moved slightly to the right, indicating that the agricultural water resources utilization efficiency in each region was gradually increasing, but it was

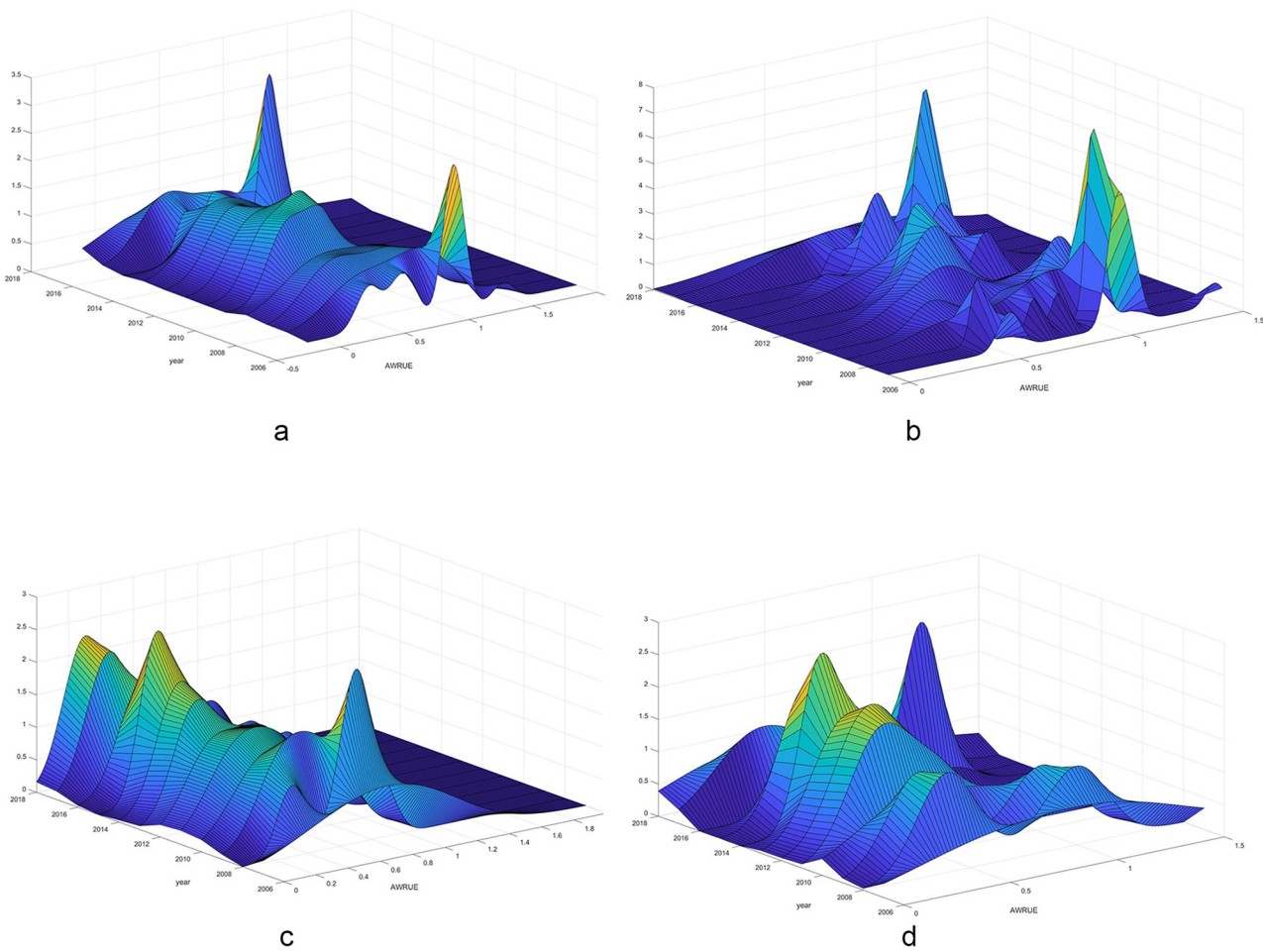

**Fig 4. Evolution of agricultural water resources utilization efficiency in different regions.**

increasing. Second, the peak height in the density map is slightly shorter, and the peak width is wider and wider, indicating that the overall gap in the agricultural water resources utilization efficiency in China is widening. This change is tortuous, but on the whole it is this trend. Before 2013, the nuclear density had a tail on the right, indicating that water resources utilization efficiency in China was at a low level before 2013. Thirdly, during the sample investigation period, the overall distribution of the peaks changes from a main peak and a small side peak to a main peak and a large side peak, indicating that there is a spatial imbalance in the development of China's agricultural water resources utilization efficiency.

**(2) Kernel density estimation of agricultural water resources utilization efficiency in the eastern region.** The Kernel density estimation of water resources utilization efficiency in the eastern region is shown in Fig 4b. It can be found that : Firstly, during the whole investigation period, the distribution of water resources utilization efficiency in the eastern provinces moved to the left, indicating that the water resources utilization efficiency in the eastern region has declined to some extent. Second, in the density map, the peak height is getting higher and the peak width is getting narrower, indicating that the gap in agricultural water resources utilization efficiency among provinces in the eastern region is narrowing. Thirdly, during the investigation period, there are multiple small side peaks in the overall distribution pattern of

the peaks, indicating that the development of agricultural water resources utilization efficiency in the eastern region is polarized and unbalanced.

**(3) Kernel density estimation of agricultural water resources utilization efficiency in central China.** The Kernel density estimation of agricultural water resources utilization efficiency in the central region is shown in Fig 4c. It can be found that : first, during the whole investigation period, the distribution of agricultural water resources utilization efficiency in the central region was basically on the left, only moving to the right slightly, indicating that the agricultural water resources utilization efficiency in the central region was at a low level and slightly improved, but there was still a large room for progress. Second, the peak in the density map is getting lower and lower, and the peak width is getting wider and wider, indicating that the gap between the agricultural water resources utilization efficiency of the provinces in the central region is becoming larger. Third, during the investigation period, the peak is still a main peak, indicating that the spatial imbalance in the central region is not particularly obvious.

**(4) Kernel density estimation of agricultural water resources utilization efficiency in the regionsregion.** The Kernel density estimation of agricultural water resources utilization efficiency in the western region is shown in Fig 4d. It can be found that : Firstly, during the whole investigation period, the distribution of water resources utilization efficiency in the western region slightly shifted to the right, indicating that the water resources utilization efficiency in the western region decreased slightly. Second, during the study period, the peaks in the density map are getting lower and lower, and the width of the peaks is getting wider and wider, indicating that the gap in water resources utilization efficiency among provinces in the western region is widening. Third, during the investigation period, the western region is always in the pattern of one main peak and multiple small side peaks, indicating that the multi-polarization trend and spatial imbalance of agricultural water resources utilization efficiency in the western region are relatively obvious.

According to the length of the sample inspection period, setting the time span of 1 year, 2 years and 3 years, agricultural water resources utilization efficiency in each province is divided into four types : low, medium-low, medium-high and high level. The transfer probability matrix obtained by Markov chain method is shown in S1 Table.

It can be seen from Table 1 that except for low-level provinces, the probability values on the diagonal lines of provinces and regions are significantly higher than those on the nondiagonal

**Table 1. Transfer probability matrix of agricultural water resources utilization efficiency in China from 2007 to 2018.**

| Time Span(year) | Type | Low | Low-Medium | Medium-High | High |
|---|---|---|---|---|---|
| 1 | Low | 0.000 | 1.000 | 0.000 | 0.000 |
| 1 | Low-medium | 0.037 | 0.840 | 0.111 | 0.012 |
| 1 | Medium-high | 0.000 | 0.171 | 0.671 | 0.158 |
| 1 | High | 0.000 | 0.018 | 0.082 | 0.901 |
| 2 | Low | 0.000 | 1.000 | 0.000 | 0.000 |
| 2 | Low-medium | 0.042 | 0.750 | 0.181 | 0.028 |
| 2 | Medium-high | 0.000 | 0.275 | 0.478 | 0.246 |
| 2 | High | 0.006 | 0.019 | 0.139 | 0.835 |
| 3 | Low | 0.000 | 1.000 | 0.000 | 0.000 |
| 3 | Low-medium | 0.061 | 0.712 | 0.167 | 0.061 |
| 3 | Medium-high | 0.000 | 0.274 | 0.500 | 0.226 |
| 3 | High | 0.000 | 0.021 | 0.142 | 0.837 |

lines, indicating that water resources utilization efficiency in all provinces and regions in China is relatively stable, showing the characteristics of club convergence. The probability of maintaining the initial state is large, and the internal liquidity is small. The situation of cross-state transfer is relatively small and the probability is low, but it still exists, indicating that water resources utilization efficiency in a province may be greatly improved in 1—3 years. With the passage of time, the probability of upward transfer to the middle and low levels in low-level areas is 100 percent, the stability of the middle and low levels is worse, and the probability of transfer to a higher level is increasing, but there is also a probability of transfer to a lower level, but the probability is small. The probability of upward transfer to a higher level in the next three years is gradually increasing. There is a possibility of transfer to the middle and low levels but there is no possibility of transfer to the low level. This shows that the probability of agricultural water resources utilization efficiency falling to a low level in the cross-stage is basically not exist. The stability of high-level areas is getting worse, and the probability of transferring to middle and low levels is increasing, that is to say, with the increase of time, the degree of club convergence is decreasing, even the provinces with higher water resources utilization efficiency should not be satisfied with the status quo or slack water management.

## Discussion

### Environmental factors affecting agricultural water resources utilization efficiency

China is a large agricultural country and a large population country. Agriculture is the basic industry of the country, and China's agricultural foundation is weak. In recent years, the existence of prominent problems such as resource waste and ecological environment destruction has seriously restricted the sustainable development of agriculture. Compared with previous studies, water resources utilization efficiency measured in this paper has a downward trend during the investigation period, which may be due to the fact that this paper measures agricultural water resources utilization efficiency, forestry, animal husbandry and fishery. When measuring water resources utilization efficiency, the author not only considers the traditional agricultural water pollution parameters (chemical oxygen demand emissions and ammonia nitrogen emissions), but also examines agricultural carbon emissions as undesirable out-puts.

In terms of water pollution control, China has begun agricultural and rural domestic sewage treatment, but rural water pollution and governance data are still ignored. Based on the availability of data, this paper only takes a small amount of pollutant indicators as undesirable outputs. It will be expanded and improved in future research. It also shows that strengthening agricultural infrastructure construction, formulating reasonable rural environmental laws and regulations and strengthening rural water environmental governance measures will effectively improve water resources utilization efficiency and promote agricultural production.

### Regional factors affecting agricultural water resources utilization efficiency

China's land area spans multiple temperature zones, with complex geographical features and regional differences in water resources distribution and rainfall. Taking into account the level of economic development, coastal location and the design of national development planning, 30 provinces, municipalities and autonomous regions in China are divided into eastern, central and western regions according to data availability. This paper further explores the contribution rate of regional differences and intra-regional differences between eastern, central and western regions to overall differences from 2007 to 2018, and faces up to the fact that regional differences and intra-regional differences exist and gradually increase, so as to play the role of

regional coordinated development and drive provinces with low efficiency of agricultural water resources from provinces with high efficiency of agricultural water resources.

## The discussion of the distribution dynamic evolution of agricultural water resources utilization efficiency in China

On a national level, there is a spatial imbalance in the development of China's agricultural water resources utilization efficiency. This shows that with the development of economy and society, the level of agricultural water resources management affects the improvement of agricultural water resources efficiency. Agriculture is no longer just "being vulnerable to any change in cilmate", but also pays attention to the scientific utilization and distribution of agricultural water resources.

From the regional level, the utilization efficiency of agricultural water resources in the eastern region has declined to a certain extent during the investigation period, and has shown a multilevel and unbalanced development. This may be due to the serious pollution and waste of agricultural water resources in the eastern region. The eastern region contains provinces and cities with multiple temperature zones in China, and the multi-polarization trend of agricultural water resources utilization efficiency is foreseeable; The efficiency of agricultural water resources in the central region has a large room for improvement, and the spatial imbalance is not particularly obvious. This may attribute to the low efficiency and waste of agricultural irrigation methods in the central region, as well as the imperfect management system of agricultural and rural water resources utilization. The utilization efficiency of agricultural water resources in the western region has increased slightly, and the trend of multi-polarization is relatively obvious. This may attribute to the continuous improvement of irrigation projects and water-saving agriculture in the western region in recent years. The western region contains a vast area, and the implementation degree of agricultural water resources management policies and the structure of agricultural industry are different, which also leads to the relatively obvious trend of multi-polarization [41, 42]. The discussion on the utilization efficiency of agricultural water resources within the region needs to be further deepened.

## Conclusion and policy recommendations

Based on the annual data of China's provinces from 2007 to 2018, this paper measures water resources utilization efficiency in China's provinces, and then uses the distribution dynamic model of Dagum Gini coefficient decomposition and expansion to examine the regional differences and distribution dynamic evolution of agricultural water resources utilization efficiency in China from 2007 to 2018 from three aspects : regional differences, shape dynamics and internal distribution liquidity. The conclusions are as follows:

First, China's water resources utilization efficiency measurement results show that China's agricultural water use level has declined substantially in recent years. The overall utilization efficiency of agricultural water resources in the eastern region is higher than that in the central and western regions, while the overall utilization efficiency of agricultural water resources in the central and western regions has little difference.

Second, according to the decomposition method of Dagum Gini coefficient, the overall regional gap shows a trend of first decreasing, then increasing, and finally decreasing, and finally increasing significantly, which is higher than that in the early stage of the investigation. Throughout the study period, the average regional gap in the eastern region was significantly lower than that in the central and western regions, the average in the western region was significantly higher than that in the eastern region, and the average in the central region was the largest. The gap between the eastern and western regions is large, and the difference between

the eastern and western regions is relatively small. During the entire investigation period, the contribution rate of inter-regional disparity is the largest, and the contribution of intra-regional disparity and hypervariable density are almost the same.

Third, Kernel density estimation results show that there is a spatial imbalance in the development of agricultural water resources utilization efficiency in China during the sample period. Water resources utilization efficiency in the eastern region has declined to some extent, and the development of agricultural water resources utilization efficiency has polarization and spatial imbalance. Water resources utilization efficiency in the central region is at a low level and slightly improved, but there is still much room for progress. The gap between provinces in the central region is widening, and the spatial imbalance in the central region is not particularly obvious. Water resources utilization efficiency in the western region has declined slightly, and the gap between provinces is widening. The multi-polarization trend and spatial imbalance of agricultural water resources utilization efficiency in the western region are more obvious.

Fourthly, the results of Markov chain analysis show that water resources utilization efficiency in China's provinces is relatively stable, showing the characteristics of club convergence. The probability of maintaining the initial state is large, and the internal liquidity is small. Over time, club convergence has declined.

In view of the above research conclusions, the following suggestions are put forward:

First, pay attention to the existence of regional differences in water resources utilization efficiency in China, in the formulation of water policy, considering the natural resource endowments, rainfall, economic and social development conditions and non-point source pollution in different provinces and regions, adjust measures to local conditions, formulate agricultural water quotas in line with local agricultural development, and develop green agriculture and water-saving agriculture in line with local planting conditions.

Second, at present, the efficiency of agricultural water use in most provinces has declined significantly, which must be paid sufficient attention to. While preventing the risk of decline in a small number of areas where the efficiency of agricultural water use has not declined, it is necessary to strengthen the scientific management of water resources and actively guide farmers to form institutional norms for scientific planting and ecological water use.

Third, in order to achieve the coordinated improvement of agricultural water resources utilization efficiency, backward provinces should learn advanced water-saving and emission reduction technologies from advanced provinces, and advanced provinces should also provide talents, knowledge and technology support to backward provinces, and promote advanced experience and measures.

## Supporting information

**S1 Fig. Agricultural water consumption in China by province for 2018.** Note:Map created using ArcGIS[10.7]. SHP data downloaded from standard map service system(http://www.resdc.cn/).
(PNG)

**S2 Fig. Agricultural water resources utilization efficiency of 30 provinces, autonomous regions, and cities in China from 2007 to 2018.** The above four charts are listed seperately according to the average water use efficiency of (a) below 0.5, (b) 0.5–0.7, (c) 0.7–0.9, and (d) 0.9 and above.
(PNG)

**S3 Fig. Diagram of decomposition results of dagum gini coefficient.**
(PNG)

**S4 Fig. Evolution of agricultural water resources utilization efficiency in different regions.**
(PNG)

**S1 Table. Transfer probability matrix of agricultural water resources utilization efficiency in China from 2007 to 2018.**
(PPTX)

**S1 Raw images.**
(PDF)

**S2 Raw images.**
(PDF)

**S3 Raw images.**
(PDF)

**S4 Raw images.**
(PDF)

## Author Contributions

**Conceptualization:** Shuya Cao.

**Data curation:** Shuya Cao.

**Formal analysis:** Shuya Cao.

**Funding acquisition:** Qian Zeng.

**Investigation:** Shuya Cao.

**Methodology:** Qian Zeng.

**Project administration:** Shuya Cao.

**Resources:** Shuya Cao.

**Software:** Jiayi H. E.

**Supervision:** Shuya Cao, Jiayi H. E.

**Validation:** Shuya Cao.

**Visualization:** Jiayi H. E.

**Writing – original draft:** Shuya Cao.

**Writing – review & editing:** Jiayi H. E.

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
