## [Decision Letter · Decision Letter 0]

13 Oct 2022

PONE-D-22-22637Regional Differences and Dynamic Evolution of Agricultural Water Resources Utilization Efficiency in ChinaPLOS ONE

Dear Dr. Cao,

Thank you for submitting your manuscript to PLOS ONE. After careful consideration, we feel that it has merit but does not fully meet PLOS ONE’s publication criteria as it currently stands. Therefore, we invite you to submit a revised version of the manuscript that addresses the points raised during the review process.

 Please submit your revised manuscript by Nov 27 2022 11:59PM. If you will need more time than this to complete your revisions, please reply to this message or contact the journal office at plosone@plos.org. Please include the following items when submitting your revised manuscript:A rebuttal letter that responds to each point raised by the academic editor and reviewer(s). You should upload this letter as a separate file labeled 'Response to Reviewers'.A marked-up copy of your manuscript that highlights changes made to the original version. You should upload this as a separate file labeled 'Revised Manuscript with Track Changes'.An unmarked version of your revised paper without tracked changes. You should upload this as a separate file labeled 'Manuscript'.

We look forward to receiving your revised manuscript.

Kind regards,

Rachata Muneepeerakul

Academic Editor

PLOS ONE

Journal Requirements:

   "This research was funded by General Project of Shaanxi Provincial Department of Science and Technology Soft Science Research Program ' Study on the Impact of Environmental Regulation on Green Technology Innovation in Shaanxi Province ' ( 2022KRM116 ), Shaanxi Social Science Fund Project ' Research on the Governance Path of Rural Water Pollution in Shaanxi Province under the Rural Revitalization Strategy ' ( 2020D025 ), Research on the driving policy of green technology innovation of enterprises in Xi ’ an under the key projects ’ and ’ double carbon ’ of Xi ’ an Social Science Planning Fund ’ ( 22JX137 ) and Key Research Project of Xi 'an University of Foreign Studies ' Governance Path and Emission Reduction Evaluation of Rural Water Pollution under Rural Re-vitalization Strategy ' ( 19XWA03 )."

   "This work was supported by “General Project of Shaanxi Provincial Depart- 507

ment of Science and Technology Soft Science Research Program ' Study on the Impact of Environmental Regulation on Green Technology Innovation in Shaanxi Province ' ( 2022KRM116 ) ”,“Shaanxi Social Science Fund Project ' Research on the Gov-ernance Path of Rural Water Pollutionin Shaanxi Province under the Rural Revitalization Strategy ' ( 2020D025 ) ”, “Research on the driving policy of green technology innovation of enterprises in Xi ’ an under the key projects ’ and ’ double carbon ’ of Xi ’ an Social Science Planning Fund ’ ( 22JX137 ) ” and “Key Research Project of Xi 'an University of Foreign Studies ' Governance Path and Emission Reduction Evaluation of Rural Water Pollution under Rural Revitalization Strategy ' ( 19XWA03 ) ”. We are indebted to the anonymous reviewers and the editor."

  "“This research was funded by General Project of Shaanxi Provincial Department of Science and Technology Soft Science Research Program ' Study on the Impact of Environmental Regulation on Green Technology Innovation in Shaanxi Province ' ( 2022KRM116 ), Shaanxi Social Science Fund Project ' Research on the Governance Path of Rural Water Pollution in Shaanxi Province under the Rural Revitalization Strategy ' ( 2020D025 ), Research on the driving policy of green technology innovation of enterprises in Xi ’ an under the key projects ’ and ’ double carbon ’ of Xi ’ an Social Science Planning Fund ’ ( 22JX137 ) and Key Research Project of Xi 'an University of Foreign Studies ' Governance Path and Emission Reduction Evaluation of Rural Water Pollution under Rural Re-vitalization Strategy ' ( 19XWA03 )."

6. We note that Figure 1 in your submission contain [map/satellite] images which may be copyrighted. All PLOS content is published under the Creative Commons Attribution License (CC BY 4.0), which means that the manuscript, images, and Supporting Information files will be freely available online, and any third party is permitted to access, download, copy, distribute, and use these materials in any way, even commercially, with proper attribution. For these reasons, we cannot publish previously copyrighted maps or satellite images created using proprietary data, such as Google software (Google Maps, Street View, and Earth). For more information, see our copyright guidelines: http://journals.plos.org/plosone/s/licenses-and-copyright.

Additional Editor Comments:

Your manuscript has been seen by two reviewers.  Both provided a set of useful comments; please make sure that you go through and address them properly in your revision.  In particular, I concur with Reviewer #2 that the novelty of this work in the context of existing literature must be explained better.  Additionally, similarities and differences between different approaches that the authors used in their analysis must be discussed more clearly.  Finally, in the future, please be more careful when you submit the same manuscript to a different journal (say, after rejection): it appears that your cover letter was addressed to Environmental Science and Pollution Research, while the template you used was for Sustainability.

Reviewers' comments:

Reviewer's Responses to Questions

**Comments to the Author**

1. Is the manuscript technically sound, and do the data support the conclusions?

Reviewer #1: Yes

Reviewer #2: Yes

2. Has the statistical analysis been performed appropriately and rigorously? 

Reviewer #1: Yes

Reviewer #2: Yes

3. Have the authors made all data underlying the findings in their manuscript fully available?

Reviewer #1: Yes

Reviewer #2: Yes

4. Is the manuscript presented in an intelligible fashion and written in standard English?

Reviewer #1: Yes

Reviewer #2: No

5. Review Comments to the Author

Reviewer #1: Overall this work shows significant effort and interesting results. I recommend this paper with minor revisions. It would benefit from a more focused presentation of the many results, along with a consolidated discussion of the context of these results. Also, while the paper was perfectly intelligible, a review for grammatical clarity would improve readability. More focus on the key results and more context for what these results represent would make this a very strong paper. Minor line revisions are presented below.

Line 80 – Define SBM acronym with first use.

Line 152 – Equation 2 or 3?

Lines 154, 161-165, and throughout – Please review terms for proper use of subscripts for clarity.

Line 245 – I’m not sure what you mean by “backward” here, please clarify.

Line 313 – These plots need a more thorough description in the caption along with larger fonts and clearly defined axes. I found the results in this section very interesting, but it would increase readability to have very clear plots that the reader can interpret for fully detailed results, allowing you to focus on the key results in the text.

Lines 333-335 – Again, this is a great point of discussion, but gets a little lost in the long format presentation of results. Integrating results and discussion together is an option, but if discussion is to be held separately, this should be relocated or reiterated there.

Reviewer #2: 1. At present, some literatures (such as Deng et al., 2016; Huang et al., 2021) have used SBM-DEA model to study water resource utilization efficiency in China. This paper also uses SBM-DEA model, so the author needs to clearly point out the innovation of this paper.

Deng, Guangyao, Lu Li, and Yanan Song. "Provincial water use efficiency measurement and factor analysis in China: based on SBM-DEA model." Ecological Indicators 69 (2016): 12-18.

Huang, Yujie, et al. "A study on the effects of regional differences on agricultural water resources utilization efficiency using super-efficiency SBM model." Scientific reports 11.1 (2021): 1-11.

2. It is recommended to list the existing literature as the basis for variable selection in Section 2.3 of the paper.

3. It is suggested to divide section 3 into 3.1 and 3.2, where 3.1 is the calculation result of water resource utilization efficiency (based on super efficiency SBM-DEA model) and 3.2 is the regional difference of water resource utilization efficiency (based on Dagum's Gini Coefficient).

4. Kernel Density Estimation and Markov Chain are both used to analyze the change trend of water resource utilization efficiency. It is suggested that the author discuss the relationship and difference between them.

5. The author is suggested to add a discussion section, which is used to compare the similarities and differences between the research results of this paper and those of the existing literature.

6. PLOS authors have the option to publish the peer review history of their article (what does this mean?). If published, this will include your full peer review and any attached files.

Reviewer #1: No

Reviewer #2: No

---

## [Author Response · Author response to Decision Letter 0]

13 Dec 2022

Journal:PLOS ONE

Date:NOV.13,2022

Dear Reviewers:g

Thank you for your letter and the reviewers’ comments on our manuscript entitled "Regional Differences and Dynamic Evolution of Agricultural Water Resources Utilization Efficiency in China" . Those comments are very helpful for revising and improving our paper, as well as the important guiding significance to other research. We have studied the comments carefully and made corrections which we hope meet with approval. The main corrections are in the manuscript and the responds to the reviewers’ comments are as follows (the replies are highlighted in blue ).

Replies to the reviewers’ comments:

Reviewer#1:

1.Line 80 - Define SBM acronym with first use.

Response：When using SBM for the first time, note the specific meaning in parentheses.It is changed to SBM (Slacks-Based Measure) in line 52.

2.Line 152 - Equation 2 or 3?

Response：It is Equation 3 in line 139.

3.Lines 154, 161-165, and throughout – Please review terms for proper use of subscripts for clarity.

Response：We have marked the subscripts correctly and make it clear.

4.Line 245 – I’m not sure what you mean by “backward” here, please clarify.

Response：“backward”is not properly used and We have replaced it with the word“lagging”.

5.Line 313 – These plots need a more thorough description in the caption along with larger fonts and clearly defined axes. I found the results in this section very interesting, but it would increase readability to have very clear plots that the reader can interpret for fully detailed results, allowing you to focus on the key results in the text.

Response：We have enlarged Figure 4 so that the axes and values can be clearly seen.

6.Lines 333-335 – Again, this is a great point of discussion, but gets a little lost in the long format presentation of results. Integrating results and discussion together is an option, but if discussion is to be held separately, this should be relocated or reiterated there.

Response：We have added a discussion section and divided it into three subsections in line392-453.

Reviewer#2:

1. At present, some literatures (such as Deng et al., 2016; Huang et al., 2021) have used SBM-DEA model to study water resource utilization efficiency in China. This paper also uses SBM-DEA model, so the author needs to clearly point out the innovation of this paper.

Deng, Guangyao, Lu Li, and Yanan Song. "Provincial water use efficiency measurement and factor analysis in China: based on SBM-DEA model." Ecological Indicators 69 (2016): 12-18.

Huang, Yujie, et al. "A study on the effects of regional differences on agricultural water resources utilization efficiency using super-efficiency SBM model." Scientific reports 11.1 (2021): 1-11.

Response：We elaborate on the innovation of the paper in line55-72.

2.It is recommended to list the existing literature as the basis for variable selection in Section 2.3 of the paper.

Response：We list the existing literature as the basis for variable selection in Section 2.3.

3.It is suggested to divide section 3 into 3.1 and 3.2, where 3.1 is the calculation result of water resource utilization efficiency (based on super efficiency SBM-DEA model) and 3.2 is the regional difference of water resource utilization efficiency (based on Dagum's Gini Coefficient).

Response：We have divided section 3(Regional Differences in Agricultural Water Resources 200

Utilization Efficiency in China) into 3.1 and 3.2,where 3.1 is the calculation result of water resource utilization efficiency (based on super efficiency SBM-DEA model) and 3.2 is the regional difference of water resource utilization efficiency (based on Dagum's Gini Coefficient).

4.Kernel Density Estimation and Markov Chain are both used to analyze the change trend of water resource utilization efficiency. It is suggested that the author discuss the relationship and difference between them.

Response：We have discussed the relationship and difference between Kernel Density Estimation and Markov Chain in line 60-72.

5.The author is suggested to add a discussion section, which is used to compare the similarities and differences between the research results of this paper and those of the existing literature.

Response：We have added a discussion section and divided it into three subsections in line 392-453.References are also attached to the discussion.

Once again, thank you very much for your constructive comments and suggestions which would help us both in English and in depth to improve the quality of the paper.Wish you a happy life!

Kind regards，

Shuya Cao

Email:caoshuya2022@163.com

Corresponding author: Shuya Cao

Email address:caoshuya2022@163.com

---

## [Decision Letter · Decision Letter 1]

7 Feb 2023

Regional Differences and Dynamic Evolution of Agricultural Water Resources Utilization Efficiency in China

PONE-D-22-22637R1

Dear Dr. Cao,

We’re pleased to inform you that your manuscript has been judged scientifically suitable for publication and will be formally accepted for publication once it meets all outstanding technical requirements.

Kind regards,

Rachata Muneepeerakul

Academic Editor

PLOS ONE

Additional Editor Comments (optional):

Reviewers' comments:

Reviewer's Responses to Questions

**Comments to the Author**

1. If the authors have adequately addressed your comments raised in a previous round of review and you feel that this manuscript is now acceptable for publication, you may indicate that here to bypass the “Comments to the Author” section, enter your conflict of interest statement in the “Confidential to Editor” section, and submit your "Accept" recommendation.

Reviewer #1: All comments have been addressed

Reviewer #2: All comments have been addressed

2. Is the manuscript technically sound, and do the data support the conclusions?

Reviewer #1: Yes

Reviewer #2: No

3. Has the statistical analysis been performed appropriately and rigorously? 

Reviewer #1: Yes

Reviewer #2: Yes

4. Have the authors made all data underlying the findings in their manuscript fully available?

Reviewer #1: Yes

Reviewer #2: Yes

5. Is the manuscript presented in an intelligible fashion and written in standard English?

Reviewer #1: Yes

Reviewer #2: Yes

6. Review Comments to the Author

Reviewer #1: Previous comments were well addressed in the revision. The manuscript is technically sound and the data supports the conclusions.

Reviewer #2: The authors revised this paper according to the comments of the reviewers, so it is recommended to accept it.

7. PLOS authors have the option to publish the peer review history of their article (what does this mean?). If published, this will include your full peer review and any attached files.

Reviewer #1: No

Reviewer #2: No

---

## [Editor Report · Acceptance letter]

17 Feb 2023

PONE-D-22-22637R1 

Regional Differences and Dynamic Evolution of Agricultural Water Resources Utilization Efficiency in China 

Dear Dr. Cao:

I'm pleased to inform you that your manuscript has been deemed suitable for publication in PLOS ONE. Congratulations! Your manuscript is now with our production department. 

Kind regards, 

on behalf of

Dr Rachata Muneepeerakul 

Academic Editor

PLOS ONE